# Short-term prophylaxis in patients with angioedema due to C1-inhibitor deficiency undergoing dental procedures: An observational study

Andrea Zanichelli[1]*, Mario Ghezzi[2☉], Ivan Santicchia[1‡], Romualdo Vacchini[1‡], Marco Cicardi[3‡], Antonella Sparaco[2‡], Girolamo Donati[2‡], Vito Ranìa[2‡], Alberto Busa[2‡]

1 Department of Biomedical and Clinical Sciences Luigi Sacco, University of Milan, ASST Fatebenefratelli Sacco, Milan, Italy, 2 Odontoiatric Unit, ASST Fatebenefratelli Sacco, Milan, Italy, 3 Department of Biomedical and Clinical Sciences Luigi Sacco, University of Milan, ICS Maugeri Department of Medicine and Rehabilitation, Milan, Italy

☉ These authors contributed equally to this work.
‡ These authors also contributed equally to this work.
* andrea.zanichelli@unimi.it

**Data Availability Statement:** All relevant data are within the manuscript and its Supporting Information files.

## Abstract

### Background

Patients affected by angioedema due to hereditary and acquired C1-inhibitor (C1-INH) deficiency (HAE and AAE, respectively) report trouble accessing dental care, due to the risk of a life-threatening oropharyngeal and laryngeal attack triggered by dental procedures.

The aim of this study was to assess the identification of hurdles in receiving dental care, and the effectiveness of short-term prophylaxis (STP) in preventing angioedema attacks. In addition, the study evaluated the impact of dental care in angioedema disease.

All patients affected by angioedema due to C1-INH deficiency who were treated in the dentistry outpatient department of ASST Fatebenefratelli Sacco hospital (Milan, Italy) between 2009 and 2017 were considered for the analysis. Data were collected from patients' records.

### Results

Twenty-nine patients were analyzed (27 with HAE and 2 with AAE). Of these, 63.0% reported that they had previously experienced hurdles in accessing dental care. Among patients with pathological oral status, at the first visit, 59.26% patients had moderate-to-severe oral disease. Seventy-five dental procedures were performed in 20 patients. Sixty procedures were preceded by STP (58 with plasma-derived C1-INH and 2 with danazol) in patients with/without long-term prophylaxis (LTP). Post-procedural attacks occurred in two patients. One HAE patient undergoing a tooth extraction without STP/LTP experienced a laryngeal attack. The other post-procedural attack occurred in an AAE patient with anti-C1-INH antibodies with STP with pdC1-INH. The angioedema disease did not worsen in any patient after dental care, but improved in four of them.

**Funding:** Publishing support and journal styling services were provided by SEEd Medical Publishers and funded by CSL Behring, Italy (https://www.cslbehring.it/). The funders had no role in study design, data collection and analysis, decision to publish, or preparation of the manuscript.

**Competing interests:** AZ received speaker/ consultancy fees and/or was a member of medical/ advisory boards for CSL Behring, Shire, and SOBI. MC received grants from Shire and personal fees from Alnylam, BioCryst, CSL Behring, Dyax, KalVista, Pharming Technologies, Shire, Sobi (Swedish Orphan Biovitrum), and ViroPharma. All the authors declare that publishing support and journal styling services for this article were funded by CSL Behring. This does not alter our adherence to PLOS ONE policies on sharing data and materials.

## Conclusions

**Most** C1-INH-HAE **patients reported hurdles in receiving dental care**. STP protects against attacks after dental procedures. Treating oral diseases results in improvement in the frequency of attacks.

## Introduction

Reduced levels of C1 esterase inhibitor (C1-INH) may be due to genetic defects in the SERPING1 gene (hereditary angioedema—HAE) or acquired deficiency (acquired angioedema—AAE) often associated with lymphoproliferative disorders. In most AAE patients, neutralizing anti-C1-INH antibodies are present [1,2]. Uncontrolled contact/kinin-systems due to C1-INH deficiency generate bradykinin, the mediator of increased vascular permeability, resulting in recurrent angioedema attacks that may affect the extremities, genitourinary tract, face, oropharynx, larynx, and abdomen [1,3].

In patients with C1-INH deficiency, dental procedures are potential triggers of angioedema attacks that may even affect the larynx [4], endangering the patient's life [5]. Since angioedema with C1-inhibitor deficiency is a rare disease, dentists are often unfamiliar with the management of attacks and do not dare to treat dental diseases in these patients. Similarly, patients are reluctant to undergo dental care because of the fear of potential attacks. As a consequence, these patients may suffer a lack of proper dental care.

The referral center for hereditary angioedema in Milano collaborates with the dentistry outpatient department to guarantee that patients affected by angioedema due to C1-INH deficiency receive proper dental care.

The aim of this study was to assess:

1. the hurdles in receiving dental care;

2. the effectiveness of short-term prophylaxis (STP) in preventing angioedema attacks;

3. the positive impact of dental care on the disease in patients with angioedema.

The latter analysis was performed in order to test the hypothesis that proper dental treatment results in an improvement in the course of the disease.

## Methods

### Ethics statements

The study was approved by the Comitato Etico Interaziendale Milano Area A with protocol number 2015/SP/253. Data collection and analysis were conducted by the patients' own physicians; therefore, patient confidentiality was well maintained. The study was conducted in accordance with the principles of the Declaration of Helsinki.

Written informed consent for using their data for this study was collected by the patients, or, if under age, their parents or legal guardians.

### Study population

The medical records of all patients affected by angioedema due to C1-INH who were diagnosed according to the criteria outlined by the Hereditary Angioedema International Working Group [1] and treated in the dentistry outpatient department of ASST Fatebenefratelli Sacco hospital (Milan, Italy) from 2009 to 2017 were considered for the analysis (see also S1 Data).

The following data were collected from patients' records: demographic information, hurdles in access to dental care, oral health at first visit, follow-up, dental procedures, type of angioedema, attack frequency, attack therapy, long- and short-term prophylaxis, and onset of attacks in the 48 hours after the procedure. Data on attack frequency were collected from angioedema records one year before and one year after the procedure. Missing data in records were collected by phone calls with patients.

In almost all patients STP consisted of the administration of plasma-derived C1-INH (pd C1-INH, Berinert®, CSL Behring, Pennsylvania, USA) one to three hours before the procedure. In one patient attenuated androgens (danazol 400mg/day) were administered every day starting five days before the procedure, and continued for two days after. In HAE and AAE patients, 1,000 IU and 1,500 IU were given, respectively.

## Endpoints

The primary endpoints were:

1.  percentage of patients encountering hurdles in the access to dental care;

2.  percentage of procedures followed by post-procedural attacks in patients receiving and nonreceiving a prophylaxis for angioedema

The secondary endpoint was the number of attacks per month in the 12 months following the procedure compared with the frequency in the year preceding the procedure.

## Severity of oral and angioedema disease

To evaluate the severity of oral pathology in our patient population, we established a score based on the evaluation of plaque, gingivitis, periodontal disease, caries, abscesses, need for tooth extractions, edentulism, odontogenic cysts, and need for implantation (Table 1).

The dental procedures performed were oral hygiene, dental fillings and root canal treatment, extractions, devitalizations, abscess care, dental prostheses, dental bridges, dental implants, treatment of odontogenic cysts, and laser excision of tongue tumors.

The severity of the course of angioedema was evaluated considering the number of attacks per month before and after dental procedures (Table 2).

## Statistical analyses

Continuous variables were reported as median and interquartile range (IQR). Categorical variables were reported as absolute frequencies and percentages. These calculations were performed by means of MS Excel 2010®.

## Results

Twenty-nine patients with angioedema due to C1-INH deficiency were visited in the dentistry department of ASST Fatebenefratelli Sacco hospital in the period considered. Fourteen were male (48.3%). The median age was 45 years (IQR = 24–53). The youngest patient was 8 years old, while the eldest was 85 years old. Twenty-seven of them were affected by C1-INH-HAE (93.1%) and two by C1-INH-AAE (6.9%).

According with records, 17/27 patients (63.0%) encountered hurdles in accessing dental care (data about two patients were missing). Two patients, 13- and 19-year-olds, were teenagers and their oral status at the first visit was judged not pathological. Among patients with pathological oral status, the majority (59.26%) had moderate-to-severe oral disease (Table 3).

**Table 1. Score to categorize the severity of oral pathology: Absent if the sum of points equals 0, mild if it is between 1 and 3, moderate 4–6, severe ≥7.**

| Pathology | Presence/number | Points |
|---|---|---|
| Plaque | No | 0 |
|  | Yes | 1 |
| Gingivitis | No | 0 |
|  | Yes | 1 |
|  | Gingival pockets or diffuse | 2 |
| Periodontal disease | No | 0 |
|  | Yes | 2 |
| Caries | 0 | 0 |
|  | 1–4 | 1 |
|  | >5 | 2 |
| Abscesses | No | 0 |
|  | Yes | 2 |
| Need for tooth extractions | 0 | 0 |
|  | 1–3 | 2 |
|  | >4 | 3 |
| Edentulism | 0 | 0 |
|  | Partial | 3 |
|  | Total | 5 |
| Odontogenic cysts | No | 0 |
|  | Yes | 2 |
| Need for implantation | No | 0 |
|  | Yes | 2 |

The most common conditions were caries (48.4% of patients), need for tooth extractions (41.9%), and edentulism (38.7%).

Nine out of twenty-nine patients had a first visit and did not undergo dental procedures in our dentistry department: two had a non-pathological status and seven had mild-to-moderate oral status, and after the first visit have turned to other dentistry services so were not considered for follow-up.

In addition, we looked for a correlation between patient age and the disease pattern, and we found that patients affected by more severe oral pathology tended to be elder, without statistical significance (p = 0.1953) (Fig 1).

Conversely, no correlations were found between age and angioedema severity.

Therefore, 20 patients (69.0%) were considered for further analyses. In this group of patients, eight were male (40.0%). The median age was 45 years (IQR = 27–61). The age range was 8–89 years, but the only underage patient (the 8-year-old one) underwent a single

**Table 2. Score to categorize the severity of angioedema course based on the number of attacks per month.**

| N. of attacks/month | Points |
|---|---|
| < 1 | 1 |
| 1 | 2 |
| 2 | 3 |
| 3 | 4 |
| ≥ 4 | 5 |

**Table 3. Oral pathological status of patients (n = 27) at the first visit, based on our severity score.**

| Pathological oral status | Score | Patients per score (n, %) | Total patients (n, %) |
|---|---|---|---|
| Mild | 1 | 3 (11.11%) | 11 (40.74%) |
| | 2 | 3 (11.11%) | |
| | 3 | 5 (18.52%) | |
| Moderate | 4 | 7 (25.92%) | 9 (33.33%) |
| | 5 | 1 (3.70%) | |
| | 6 | 1 (3.70%) | |
| Severe | 7 | 4 (14.81%) | 7 (25.92%) |
| | 8 | 3 (11.11%) | |

Median score was 4 (IQR = 2–6).

procedure, that was an oral hygiene. Eighteen patients (90.0%) were affected by C1-INH-HAE and two (10.0%) by C1-INH-AAE.

Overall, ten patients (50%), affected by C1-INH-HAE, were on LTP: nine with attenuated androgens and one with pdC1-INH administered intravenously. Both the patients with C1-INH-AAE were not on LTP.

Seventy-five procedures were performed on twenty patients (Table 4).

Table 4 shows types of procedure and the preceding prophylaxis.

The majority of procedures were undertaken after prophylaxis. In particular, 60 procedures (80%) were preceded by STP in patients with/without LTP. Thirteen procedures were undertaken without STP in patients regularly taking LTP.

Only two procedures in two hereditary angioedema patients were not preceded by any type of prophylaxis: one was oral hygiene, while the other was performed on a patient who had an angioedema attack within 48 hours of the dental extraction.

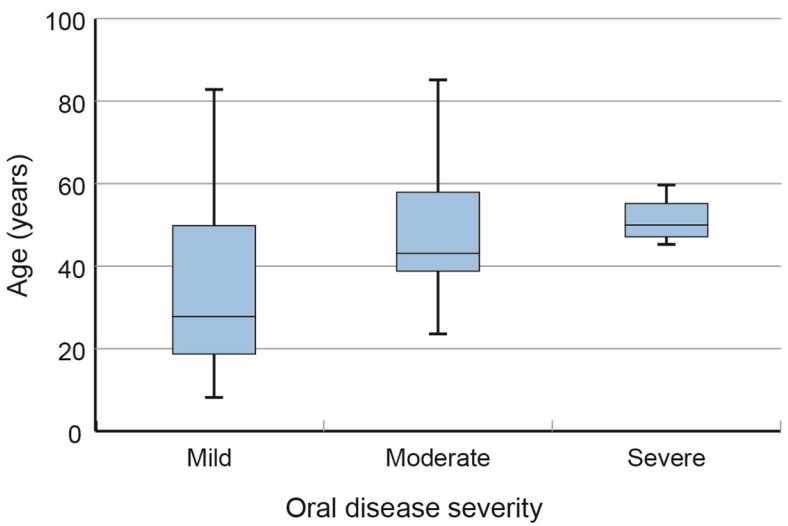

**Fig 1. Boxplot showing age distribution according to oral disease severity.** The bold lines represent the median values, the boxes indicate the interquartile range, and the whiskers represent the minimum and maximum values. Please, refer to Table 1 for the score used to categorize the severity of oral pathology.

**Table 4. Type of procedures and of prophylaxis undertaken (n = 20).**

| Type of procedure | Total (n.) | STP only (n.) | LTP only (n.) | STP and LTP (n.) | No STP, no LTP (n.) |
|---|---|---|---|---|---|
| Dental fillings and root canal treatments | 33 | 14 | 6 | 13 | 0 |
| Extractions | 20 | 7 | 1 | 11 | 1 |
| Oral hygiene | 10 | 4 | 4 | 1 | 1 |
| Dental prostheses | 4 | 0 | 0 | 4 | 0 |
| Dental implants | 2 | 0 | 0 | 2 | 0 |
| Dental bridges | 2 | 0 | 2 | 0 | 0 |
| Abscess care | 1 | 0 | 0 | 1 | 0 |
| Devitalizations | 1 | 0 | 0 | 1 | 0 |
| Treatment of odontogenic cysts | 1 | 1 | 0 | 0 | 0 |
| Laser excision of tongue tumors | 1 | 1 | 0 | 0 | 0 |

LTP, long-term prophylaxis; STP, short-term prophylaxis.

An attack after the extraction of one tooth occurred in an AAE patient not in LTP who took pdC1-INH as STP.

Data about the frequency of attacks were available in 19/20 patients.

In four patients the frequency of attacks per month in the year after the procedure was reduced compared to the frequency of attacks per month in the year before; in all other patients it remained unchanged. None experienced an increase in the number of attacks after the procedures (Fig 2).

## Discussion

In this study, the majority of patients with angioedema (63.0%) reported that they had encountered hurdles in accessing dental care. Often dentists are not aware of this disease and of the correct management of patients with angioedema undergoing dental procedures. Due to the risk of post-procedural attacks, dentists dare not treat these patients. As suggested by Forrest

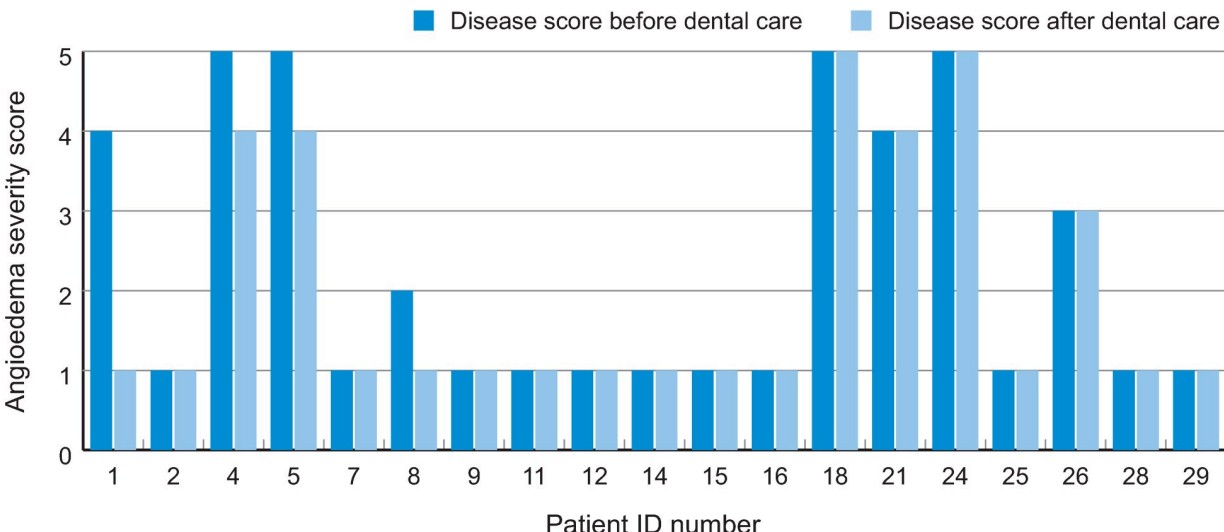

**Fig 2. Angioedema severity before and after dental procedures (refer to our score in Table 2).**

and colleagues [5], dental practitioners should be informed about this pathology, even if rare, and should add a specific question about previous swelling episodes in their habitual questionnaire for anamnestic purpose.

Difficulties in accessing dental care can result in a poor oral condition. In particular, 59.25% of patients recruited in this study had a moderate-to-high score for the severity of oral pathology. It is well known that infections may trigger angioedema attacks [6,7]. An observational study carried out by the group of Farkas [8] investigated the trigger factors in 27 patients affected by C1-INH-HAE. Patients recorded the occurrence of potential trigger factors every day for 7 months, whether or not they experienced an attack. The likelihood of angioedema attack associated with infection was 38%.

Other studies demonstrated that the eradication of *Helicobacter pilori* infection in C1-INH-HAE patients was effective in reducing the frequency of attacks [9,10].

Since the need for dental procedures is indicative of infections localized in the oral cavity, improving dental care may be a useful strategy to reduce the frequency of angioedema attacks [11]. In this study, treating C1-INH-HAE patients with severe oral pathology had a positive impact on the course of the disease: in 20% of patients the frequency of attacks was reduced after dental care. We eliminated a possible trigger of angioedema attack, but, as it is not possible to eradicate the underlying pathology, we did not expect an improvement in all the patients. Among those who improved, one had mild, two had moderate, and one had severe oral pathology. The average oral pathology score was slightly worse in this subpopulation compared with the other patients (4.75 vs 4). This finding indicates that the subpopulation who benefits most from dental procedures has more severe oral pathology.

As stated by Longhurst [12] and the Italian guidelines on the diagnostic and therapeutic management of C1-INH-HAE [13], STP should be considered in these patients before dental procedures. Most evidence is available for pdC1-INH as STP. Danazol, in case no other drugs are available, may be used.

STP was effective in preventing post-procedural attacks. All C1-INH-HAE patients with STP, in the presence or in the absence of LTP, did not suffer attacks. Further confirmation comes from the fact that one patient had an attack just when not protected by STP.

In fact, a patient with C1-INH-HAE without LTP, informed of the risk of the possible attack, decided not to be treated with STP before a tooth extraction. This patient had a post-procedural attack, located in the oropharynx, that was treated in the Emergency Department (ED) with pdC1-INH, with symptoms resolution. When this patient underwent another procedure (tooth extraction), he did not manifest any attack after receiving STP with pdC1-INH.

The effectiveness of STP with pdC1-INH was shown by the retrospective study of Bork [4], which analyzed clinical records of C1-INH-HAE patients undergoing tooth extractions. Angioedema attacks occurred in 21.5% (124/577) of tooth extractions without STP versus 12.5% (16/128) of tooth extractions with STP, highlighting a 41.9% reduction in angioedema attacks when using pdC1-INH before the procedure ($p < 0.05$).

In the analysis of Bork, many attacks occurred within 12 hours after tooth extractions, making the night following the dental procedure the most dangerous moment for attack onset. Finally, the retrospective analysis of Bork detected a significant dose-response effect (21.5% attacks without prophylaxis, 16.0% with 500 IU, and 7.5% with 1,000 IU). A trend toward a dose-response effect of pdC1-INH is also suggested by Magerl and coauthors in the analysis of Berinert Registry, collecting data from 30 US and 7 European centers between 2010 and 2014 [14].

In our cohort of C1-INH-HAE patients, pdC1-INH (Berinert®) was used as STP at a fixed dose of 1,000 IU.

Another retrospective analysis [6] confirmed the effectiveness of pdC1-INH as STP, highlighting also a superiority if compared with danazol and tranexamic acid. Invasive medical

interventions, including dental procedures, before and after the diagnosis of C1-INH-HAE were analyzed in order to compare the onset of attacks with and without STP. The analysis detected a significant reduction in the number of edematous episodes when using a STP (39/89 vs 3/55, i.e. 43.8% vs 5.4%). In our cohort, only one patient received danazol as STP, thus a comparison with STP with pdC1-INH is not possible.

In our study, in C1-INH-AAE patients higher doses of pdC1-INH were used for STP, since it is known that in these patients the catabolism of C1-INH is faster [2].

One patient affected by C1-INH-AAE without anti-C1-INH antibodies underwent five dental fillings and root canal treatments, two oral hygiene procedures, and one laser excision of two tongue tumors with STP, and had no attacks.

Another patient affected by C1-INH-AAE with anti-C1-INH antibodies underwent a tooth extraction and manifested a post-procedural attack within 24 hours of the dental procedure despite STP (pdC1-INH). The patient was not on LTP. The attack, located in the oropharyngeal tract, was severe and treated in ED.

STP with pdC1-INH seems to be less efficacious in patients affected by AAE with anti-C1-INH antibodies. There is no evidence from randomized clinical trials that antifibrinolytic agents are effective for STP in patients with C1-INH-AAE. In addition, antifibrinolytic agents are contraindicated in some categories of patients, such as those receiving antiplatelet or anticoagulant therapy and those with procoagulant condition, such as the patient with AAE with anti-C1-INH antibodies enrolled in our study. For this reason, we used STP with pdC1-INH at higher doses in this patient.

In our study, the triggering procedure for angioedema attacks was tooth extraction. Bork and colleagues also reported that tooth extraction was the most common triggering factor in the head region in C1-INH-HAE patients [4]. Although dentists consider a tooth extraction in general population as a routine procedure, in patients with C1-INH deficiency it is a common trigger for an attack. Therefore, STP is strongly recommended in these patients who are undergoing tooth extraction.

This study has some limitations. First, the sample size is low, but the rarity of the disease makes it difficult to study a higher number of patients. An international register for C1-INH-HAE collecting data from high number of patients may overcome these issues. Such a register will allow physicians to better evaluate the effectiveness of the drugs used in this pathology, even for STP.

Second, the score used to evaluate the severity of oral disease in this study was developed by the authors since in the literature there is no consensus on this topic. Therefore, data concerning oral severity assessed at the first visit of this study cannot be compared with those found in the literature.

Third, the data collection of the study was carried out through information present in the patients' records. As well known, secondary data are less affordable than data coming from randomized clinical trials. Further interventional studies, even though difficult to perform owing to the rarity of the disease, are needed in order to confirm these conclusions.

Fourth, pediatric and adult patients were considered together. However, among those who underwent dental procedures, only one patient was underage and received just an oral hygiene.

## Conclusions

In conclusion, this analysis revealed that most patients with angioedema due to C1-INH deficiency encountered hurdles in receiving dental care because dentists are not familiar with this disease and its treatments.

STP with pdC1-INH was effective in preventing post-procedural attacks. However, rescue therapy should always be available for patients undergoing dental procedures.

A considerable percentage of patients (21.05%) experienced a reduction in the frequency of angioedema attacks after receiving dental care. This highlights the importance of treating oral pathologies in patients affected by angioedema due to C1-INH deficiency.

Taking into account our results and other data in the literature, we recommend considering tooth extractions as posing a high risk of attack in these patients and therefore advise the use of STP before these procedures.

## Supporting information

**S1 Data.**
(XLSX)

## Acknowledgments

We acknowledge Laura Fascio Pecetto from SE*Ed* Medical Publishers for the medical writing service.

## Author Contributions

**Conceptualization:** Andrea Zanichelli, Mario Ghezzi.

**Data curation:** Andrea Zanichelli, Mario Ghezzi, Ivan Santicchia.

**Formal analysis:** Ivan Santicchia.

**Investigation:** Andrea Zanichelli, Mario Ghezzi, Ivan Santicchia, Romualdo Vacchini, Marco Cicardi, Antonella Sparaco, Girolamo Donati, Vito Ranìa, Alberto Busa.

**Methodology:** Andrea Zanichelli, Mario Ghezzi.

**Resources:** Andrea Zanichelli, Mario Ghezzi.

**Supervision:** Andrea Zanichelli, Mario Ghezzi.

**Writing – review & editing:** Andrea Zanichelli, Mario Ghezzi.

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
