## [Decision Letter · Decision Letter 0]

27 Nov 2019

PONE-D-19-20853

Short-term prophylaxis in patients with angioedema due to C1-inhibitor deficiency undergoing dental procedures: an observational study

PLOS ONE

Dear Dr zanichelli,

Thank you for submitting your manuscript to PLOS ONE. After careful consideration, we feel that it has merit but does not fully meet PLOS ONE’s publication criteria as it currently stands. Therefore, we invite you to submit a revised version of the manuscript that addresses the points raised during the review process.

We would appreciate receiving your revised manuscript by Jan 11 2020 11:59PM. To enhance the reproducibility of your results, we recommend that if applicable you deposit your laboratory protocols in protocols.io, where a protocol can be assigned its own identifier (DOI) such that it can be cited independently in the future. For instructions see: http://journals.plos.org/plosone/s/submission-guidelines#loc-laboratory-protocols

We look forward to receiving your revised manuscript.

Kind regards,

Girijesh Kumar Patel, PhD

Academic Editor

PLOS ONE

Journal Requirements:

'Publishing support and journal styling services were provided by SEEd Medical Publishers and funded by CSL Behring, Italy (https://www.cslbehring.it/). The funders had no role in study design, data collection and analysis, decision to publish, or preparation of the manuscript'

We note that you received funding from a commercial source: CSL Behring

Additional Editor Comments (if provided):

Reviewers' comments:

Reviewer's Responses to Questions

**Comments to the Author**

1. Is the manuscript technically sound, and do the data support the conclusions?

Reviewer #1: Partly

Reviewer #2: Yes

2. Has the statistical analysis been performed appropriately and rigorously? 

Reviewer #1: I Don't Know

Reviewer #2: Yes

3. Have the authors made all data underlying the findings in their manuscript fully available?

Reviewer #1: Yes

Reviewer #2: Yes

4. Is the manuscript presented in an intelligible fashion and written in standard English?

Reviewer #1: Yes

Reviewer #2: Yes

5. Review Comments to the Author

Reviewer #1: I think this paper can only be accepted if the author's add and answer the following points in the current submission. It needs more statistical analysis to be done with statistical software packeges.

1. Kindly add co-relation of patient age with the disease pattern. In current article patients age 8-85 years are added, which is a wide range.

2. Add some more statistical data analysis as well as some statistical graph in this article.

3. Add some more discussion to correlate the immunological effect of the disease process as well as therapeutic approach.

4. Kindly add some more current relevant references.

5. Add some pictures to show the Angioderama and clarify the scoring process with more supporting images in the current population of the patient.

Reviewer #2: The authors present data from a study on the management of dental procedures in patients with HAE or AAE due to C1-Inhibitor deficiency. It is adequately written and the conclusions presented are appropriately supported by the data. Although most of the conclusions reached by the authors are not novel, they do provide experimental support to previously reported observations with important implications for the management of these patients. Because of dealing with a rare pathology, the relatively small number of individuals recruited for this study is not to be considered a substantial limitation in its design.

Minor points:

- Lines 175-176: The manuscript states that “….Most patients recruited in this study had a moderate-to-high score for the severity of oral pathology”. However, according to Table 1, most of the patients studied (20/27) had a Mild to Moderate phenotype.

- Lines 179-180: The amelioration of the patients’ HAE course after dental care is a reasonable conclusion. Though, significant improvement of HAE was observed in as little as 20% of the studied cases. Wouldn’t it be expectable to find a general improvement in the cohort after dental management? Were these 4 patients showing amelioration of HAE those exhibiting worse oral pathological scores?; that is, is there any correlation between the oral disease score and the beneficial effect of dental care on HAE?

Can the authors speculate in the discussion section on the hypothetical causes of the lack of response in an 80% of cases?

- Line 223: Previous studies have also shown that pdC1INH prophylaxis is less effective for AAE-C1INH as compared to HAE-C1INH (reviewed for example in Cicardi et al, 2014). This observation is coherent with the increased C1INH catabolism characteristic of AAE-C1INH patients and may be influenced by the presence or absence of anti-C1INH autoantibodies in the patient, as suggested by the authors’ data.

However, most experts recommend antifibrinolytic agents for the prophylaxis of AAE-C1INH; why were the two AAE-C1INH patients receiving pdC1INH instead? The reason of this treatment choice should be clear in the manuscript.

6. PLOS authors have the option to publish the peer review history of their article (what does this mean?). If published, this will include your full peer review and any attached files.

Reviewer #1: Yes: Dr. Moumita Roy

Reviewer #2: No

---

## [Author Response · Author response to Decision Letter 0]

9 Dec 2019

REVIEWER

Reviewer #1: I think this paper can only be accepted if the author's add and answer the following points in the current submission. It needs more statistical analysis to be done with statistical software packeges.

1. Kindly add co-relation of patient age with the disease pattern. In current article patients age 8-85 years are added, which is a wide range.

2. Add some more statistical data analysis as well as some statistical graph in this article.

AUTHORS

As required by Reviewer #1, we performed more statistical analyses. We investigated the possible correlation between age and oral disease severity and between age and angioedema severity. We added two paragraphs and one figure in the Results section.

REVIEWER

3. Add some more discussion to correlate the immunological effect of the disease process as well as therapeutic approach.

4. Kindly add some more current relevant references.

AUTHORS

We added a more thorough explanation about the relationship between infections and occurrence of angioedema attacks in the Discussion section and we circumstantiated these data with appropriate bibliographic references.

REVIEWER

5. Add some pictures to show the Angioderama and clarify the scoring process with more supporting images in the current population of the patient.

AUTHORS

We showed our scoring systems, adding Table 1 and Table 2 in the Methods section (see “Severity of oral and angioedema disease” paragraph).

REVIEWER

Reviewer #2: The authors present data from a study on the management of dental procedures in patients with HAE or AAE due to C1-Inhibitor deficiency. It is adequately written and the conclusions presented are appropriately supported by the data. Although most of the conclusions reached by the authors are not novel, they do provide experimental support to previously reported observations with important implications for the management of these patients. Because of dealing with a rare pathology, the relatively small number of individuals recruited for this study is not to be considered a substantial limitation in its design.

AUTHORS

We thank the Reviewer for appreciating this work.

REVIEWER

Minor points:

- Lines 175-176: The manuscript states that “….Most patients recruited in this study had a moderate-to-high score for the severity of oral pathology”. However, according to Table 1, most of the patients studied (20/27) had a Mild to Moderate phenotype.

AUTHORS

The intended meaning was that the sum of patients with moderate and severe oral pathology is greater than 50%. In order to avoid misunderstandings, we changed the sentence indicating the precise percentage (59.25%).

REVIEWER

- Lines 179-180: The amelioration of the patients’ HAE course after dental care is a reasonable conclusion. Though, significant improvement of HAE was observed in as little as 20% of the studied cases. Wouldn’t it be expectable to find a general improvement in the cohort after dental management? Were these 4 patients showing amelioration of HAE those exhibiting worse oral pathological scores?; that is, is there any correlation between the oral disease score and the beneficial effect of dental care on HAE?

Can the authors speculate in the discussion section on the hypothetical causes of the lack of response in an 80% of cases?

AUTHORS

We thank the Reviewer, as our manuscript would certainly benefit from further deepening of this topic. 

We eliminated a possible trigger of angioedema attack, but, as it not possible to eradicate the underlying pathology, we did not expect an improvement in all the patients. No one worsened and 4 improved: we added in the Discussion section a thorough analysis of the severity of oral pathology.

The average oral pathology score was slightly worse in the subpopulation who experienced an amelioration compared with the other patients (4.75 vs 4). This finding indicates that the subpopulation who benefits most from dental procedures has more severe oral pathology.

REVIEWER

- Line 223: Previous studies have also shown that pdC1INH prophylaxis is less effective for AAE-C1INH as compared to HAE-C1INH (reviewed for example in Cicardi et al, 2014). This observation is coherent with the increased C1INH catabolism characteristic of AAE-C1INH patients and may be influenced by the presence or absence of anti-C1INH autoantibodies in the patient, as suggested by the authors’ data.

However, most experts recommend antifibrinolytic agents for the prophylaxis of AAE-C1INH; why were the two AAE-C1INH patients receiving pdC1INH instead? The reason of this treatment choice should be clear in the manuscript.

AUTHORS

We thank the Reviewer for this comment. There is no evidence from randomized clinical trials that antifibrinolytic agents are effective for STP in AAE-C1INH. These agents are contraindicated in some categories of patient, such as those receiving antiplatelet or anticoagulant therapy and those with procoagulant condition. This was the case of our patients and the reason why we chose pdC1INH for STP.

We clarified the reason of the treatment choice in the manuscript in the Discussion section.

---

## [Decision Letter · Decision Letter 1]

24 Feb 2020

Short-term prophylaxis in patients with angioedema due to C1-inhibitor deficiency undergoing dental procedures: an observational study

PONE-D-19-20853R1

Dear Dr.  Zanichelli,

We are pleased to inform you that your manuscript has been judged scientifically suitable for publication and will be formally accepted for publication once it complies with all outstanding technical requirements.

With kind regards,

Girijesh Kumar Patel, PhD

Academic Editor

PLOS ONE

Additional Editor Comments (optional):

Reviewers' comments:

Reviewer's Responses to Questions

**Comments to the Author**

1. If the authors have adequately addressed your comments raised in a previous round of review and you feel that this manuscript is now acceptable for publication, you may indicate that here to bypass the “Comments to the Author” section, enter your conflict of interest statement in the “Confidential to Editor” section, and submit your "Accept" recommendation.

Reviewer #2: All comments have been addressed

Reviewer #3: (No Response)

2. Is the manuscript technically sound, and do the data support the conclusions?

Reviewer #2: Yes

Reviewer #3: (No Response)

3. Has the statistical analysis been performed appropriately and rigorously? 

Reviewer #2: Yes

Reviewer #3: (No Response)

4. Have the authors made all data underlying the findings in their manuscript fully available?

Reviewer #2: Yes

Reviewer #3: (No Response)

5. Is the manuscript presented in an intelligible fashion and written in standard English?

Reviewer #2: Yes

Reviewer #3: (No Response)

6. Review Comments to the Author

Reviewer #2: The authors have succesfully addressed all the questions and comments. No further reviewing is required on my part.

Reviewer #3: Your raising of awareness with regards to a highly neglected disease in Dentistry is a rewarding addition to health and science. Following past reviewer comments has greatly enhanced the content of this paper and made it more valuable, however, there are a few minor corrections (added as comments) I have pointed out on the context of your PDF document article (provided as an attachment) that I recommend you amend as a final aspect.

7. PLOS authors have the option to publish the peer review history of their article (what does this mean?). If published, this will include your full peer review and any attached files.

Reviewer #2: Yes: Alberto López Lera

Reviewer #3: No

---

## [Editor Report · Acceptance letter]

28 Feb 2020

PONE-D-19-20853R1 

Short-term prophylaxis in patients with angioedema due to C1-inhibitor deficiency undergoing dental procedures: an observational study 

Dear Dr. zanichelli:

I am pleased to inform you that your manuscript has been deemed suitable for publication in PLOS ONE. Congratulations! Your manuscript is now with our production department. 

With kind regards,

on behalf of

Dr. Girijesh Kumar Patel 

Academic Editor

PLOS ONE